



# Observation of Kelvin-Helmholtz Instabilities and gravity waves in the summer mesopause above Andenes in Northern Norway

Gunter Stober[1], Svenja Sommer[1,*], Carsten Schult[1], Ralph Latteck[1], and Jorge L. Chau[1]

[1]Leibniz-Institute of Atmospheric Physics at the University Rostock, Schlossstr. 6, 18225 Kühlungsborn, Germany.
[*]now at: Fraunhofer Institute for High Frequency Physics and Radar Techniques, Fraunhoferstr. 20, 53343 Wachtberg

*Correspondence to:* Gunter Stober (stober@iap-kborn.de)

**Abstract.** We present observations obtained with the Middle Atmosphere Alomar Radar System (MAARSY) to investigate short period wave-like features using polar mesospheric summer echoes (PMSE) as tracer for the neutral dynamics. We conducted a multi-beam experiment including 67 different beam directions during a 9-day campaign in June 2013. We identified two Kelvin Helmholtz Instability (KHI) events from the signal morphology of PMSE. The MAARSY observations are complemented by collocated meteor radar wind data to determine the mesoscale gravity wave activity and the vertical structure of the wind field above the PMSE. The KHIs occurred in a strong shear flow with Richardson numbers $Ri < 0.25$. In addition, we observed 15 wave-like events in our MAARSY multi-beam observations applying a sophisticated decomposition of the radial velocity measurements using Volume Velocity Processing. We retrieved the horizontal wavelength, intrinsic frequency, propagation direction and phase speed from the horizontally resolved wind variability for 15 events. These events showed horizontal wavelengths between 20-40 km, vertical wavelengths between 5-10 km and rather high intrinsic phase speeds between 45-85 m/s with intrinsic periods of 5-10 minutes.

## 1 Introduction

The middle atmosphere is a highly variable atmospheric region driven by a variety of waves. In particular, the dynamics of the Mesosphere/Lower Thermosphere (MLT) region is characterized by waves spanning various temporal and spatial scales e.g., Planetary Waves (PW), tides, and Gravity Waves (GW). Our current knowledge about the energy dissipation of breaking mesoscale GW at the MLT is limited due to the lack of continuous high resolution (spatial and temporal) temperature and wind measurements at these altitudes.

Optical observations, such as lidar or airglow imagers depend on the weather conditions (cloud free conditions) and are often restricted to nighttime measurements only. Airglow imagers (e.g., Hecht et al., 2000, 2005, 2007; Suzuki et al., 2013; Wüst et al., 2017) and the Mesospheric Temperature Mapper (Taylor et al., 2007) have the ability to resolve the horizontal structure in the field of view as well as to obtain information about the temporal evolution of mesoscale GW or wave like structures, often called ripples (Hecht et al., 2007). These ripples are excited when mesoscale GW break and dissipate their energy and momentum.

The nature of these ripple structures is associated to either convective or dynamical instabilities. Dynamical instabilities evolve



often into KHI, whereas the convective instability has its origin in superadiabatic temperature gradients (Hecht et al., 2005). Airglow observations as well as models (Horinouchi et al., 2002) suggest that KHI can generate secondary instabilities of convective nature. Nonetheless, the optical measurements are limited to nighttime and cloud free conditions. In addition, airglow observations are lacking the exact altitude information.

Although observations of KHIs have been reported previously using VHF radars, we try to identify such events using polar mesospheric summer echoes (PMSE) as a tracer. There are several studies of KHIs from radar observations in the troposphere (e.g., Klostermeyer and Rüster, 1980, 1981; Fukao et al., 2011, and references therein) or in the equatorial mesosphere (e.g., Lehmacher et al., 2007, 2009, and references therein). Optical observations of NLC (e.g., Demissie et al., 2014; Baumgarten and Fritts, 2014; Fritts et al., 2014, and many other), which are closely related to PMSE, show that KHI occur

rather frequently in the summer mesopause at polar latitudes and, hence, might also be seen using PMSE as a tracer.

We present in this paper measurements under daylight conditions with a high spatial and temporal resolution using a multi-beam radar experiment. The observations were conducted with the Middle Atmosphere Alomar Radar System (MAARSY) in Northern Norway (69.30° N, 16.04°E) during summer 2013. PMSE are a common phenomenon at this latitude and are suitable tracers of neutral dynamics (e.g., Chilson et al., 2002; Stober et al., 2013, and references therein).

KHI are investigated from the signal morphology of the PMSE as well as from the obtained Doppler measurements. Our Doppler velocity measurements permit to determine the amplitude of the instability and to estimate the characteristic scales during a strong shear flow. The PMSE wind observations are complemented by meteor radar winds in order to access the mean winds above and below the PMSE layer and to estimate the mesoscale stability computing the Richardson number (Ri). In the second part of the paper we investigate 15 events with wave-like features using the imaging capabilities of the radar system to

obtain horizontally resolved radial velocity images, which are analyzed with respect to the propagation direction, horizontal wavelength and phase speed (Stober et al., 2013).

The manuscript is structured as follows. A short summary of the technical details of MAARSY as well as the experiments are presented in section 2 including also a brief description of the Andenes meteor radar. The wind analysis is outlined in section 3. Section 4 contains a description of the analysis of two gravity wave induced KHI events seen in the morphology of a PMSE.

In Section 5 we review the gravity wave analysis from horizontally resolved radial velocities and present the obtained GW properties (observed and intrinsic phase speed, observed and intrinsic period, horizontal and vertical wavelength). Our results are discussed and related to other observations in section 6. Finally, we summarize and conclude our results in section 7.

## 2 Experimental Setup

MAARSY is located at the Northern Norwegian island of Andøya (69.30° N, 16.04°E). The system operates within the VHF

band at 53.5 MHz. The radar employs an active phased array consisting of 433 individual antennas. Each antenna is connected to its own transceiver module, which is freely adjustable in phase, power and frequency (within the assigned 2 MHz bandwidth around the carrier frequency). MAARSY has a peak power of 866 kW and a beam width of 3.6°. The beam is freely steerable within off-zenith angles up to 35° without generating grating lobes. A more detailed description of the radar is given in





Latteck et al. (2012) and an overview of wind field analysis using multi-beam experiments can be found in Stober et al. (2013). In summer 2013, MAARSY conducted several multi-beam experiments to provide systematic scans of the horizontal structure of PMSE using 67 unique beam pointing directions. The experiments were optimized to ensure a horizontal coverage of 80 km in diameter while keeping a fast enough sampling to obtain reliable Doppler measurements. A complete scan of the observation volume consisted of 4 experiments with 17 beams each. Each experiment did contain the vertical and 16 oblique beams. Figure 1 shows the beam positions for the complete sequence as well as for each experiment as a projection above the North Norwegian shore line (black lines). The red circles denote the diameter of the illuminated area assuming a $3.6°$ beam width.

The total time resolution between successive scans with the multi-beam experiments was 3 minutes 35 seconds. The shortest GW period that could exist at the PMSE altitude is given by the Brunt-Väisälä period, which is approximately 4 minutes at the summer polar mesopause. Given that the vertical beam is sampled at a higher temporal resolution (it is included in each experiment), it is possible to resolve even higher periodicities of approximately one minute. In so far, the spatial and temporal resolution of the multi-beam observations is sufficient to resolve short period wave-like features. The images derived from the multi-beam experiments permit to directly access the intrinsic GW properties similar to airglow observations.

## 3  Data analysis

A summary of the experiment configuration is given in Table 6. We analyzed the recorded raw data with regard to the signal to noise ratio (SNR), the radial velocity and the spectral width using 4 incoherent integrations. Further, we obtain the statistical uncertainties from a truncated Gaussian fit to the spectra. The fitting routine is based upon the concept presented in Kudeki et al. (1999); Sheth et al. (2006); Chau and Kudeki (2006). This spectral Gaussian fitting takes the effects of the rectangular window and the temporal sampling into account.

In Figure 6 we show a contour plot of the radial velocities as well as their associated statistical uncertainties. We removed potential meteors to avoid a contamination of the measurements. Meteors are removed by checking all data points with an $SNR > -7dB$, whether there are other adjacent points, whether they also show a SNR larger than our noise floor (approximately $SNR < -7.5dB$). If there is only one point with an enhanced SNR and all surrounding ones are smaller or comparable to the noise floor $\pm 0.2dB$, we consider this measurement to be contaminated by a meteor. Further, we suppressed a potential side lobe contamination at the edges of the PMSE layer by using the interferometry and assumed consistency in the vertical profile of the radial velocities. If there is a jump in the vertical profile of more than 6 m/s from the core region towards the edges between adjacent pixels, we removed these measurements.

The lower panels in Figure 6 we show a histogram of statistical uncertainties of the radial velocity and a SNR vs. radial statistical uncertainty scatter plot. The histogram peaks at an statistical uncertainty of 0.17 m/s and has a median of about 0.56 m/s. We truncated the contour plot color scale at 5 m/s as the histogram shows almost no radial velocity uncertainties larger than 5 m/s. The SNR vs. statistical uncertainty radial velocity scatter plot further visualizes the L-shape, which means that large errors are often associated by low SNR measurements, as it is expected.



MAARSY has a multi-channel receiver system, which is used for Coherent Radar Imaging (CRI) (e.g., Woodman, 1997; Chilson et al., 2002; Sommer et al., 2014). When PMSE have relative steep gradients at the edges of the layer, the CRI technique is useful to correct for beam filling effects leading to differences between the nominal beam pointing direction and the strongest returned signal (Sommer et al., 2014). This is of particular relevance for the oblique beams with zenith distances of more than $10°$ as there could be a deviation of several degrees from the nominal beam pointing direction causing substantial errors in the derived horizontal wind velocities and altitude errors of up to 2 km (Sommer et al., 2014, 2016).

Further, we use wind observations from a collocated meteor radar. The system operates at 32.55 MHz and has a peak power of 30 kW. The radar employs a crossed dipole antenna for transmission and five crossed dipole antennas for reception (Jones et al., 1998). The radar detects meteors within a diameter of 600 km. During the summer months we observe between 15000 to 20000 meteors per day.

Winds presented in this study are computed using a full error propagation of the statistical uncertainties from the radial velocity measurements and are based on a new retrieval technique described in Stober et al. (2017). These wind estimates have been extensively compared and validated in McCormack et al. (2017) and Wilhelm et al. (2017). Here we describe the key features of the developed wind retrieval algorithm. The starting point of the retrieval is the so called All-sky fit for both data sets, which can be considered as a more general DBS (Doppler-Beam Swinging) analysis (Hocking et al., 2001). The advantage of this approach is that we can use an arbitrary number of measurements (at least three) at different positions. We additionally implemented a regularization in time and altitude to retrieve a reliable wind estimate using at least 4 meteors. The winds are obtained solving the radial wind equation iteratively to ensure a proper error propagation due to the statistical uncertainty of the radial wind velocity measurement, and the pointing directions in azimuth and zenith. Typically we need 5 iterations until we achieve convergence. Typical errors in our obtained winds are in the order of less than 1 m/s for MAARSY and 1-10 m/s for the meteor radar. The largest uncertainties occur at the upper and lower boundaries of the observed altitudes.

The multi-beam experiments are also appropriate to apply more sophisticated wind analysis methods such as the Velocity Azimuth Display (VAD) (Browning and Wexler, 1968) or the Volume Velocity Processing (VVP) (Waldteufel and Corbin, 1979). If radial velocities are not available for all beam directions due to the patchy PMSE structure it turns out that the VVP is more suitable and robust (Stober et al., 2013). In addition, the benefit of the VVP technique is to access higher order kinematic terms such as horizontal divergence, stretching and shearing deformation in the wind field. As we show in the second part of the paper, VVP allows to decompose the wind field into mean winds, mesoscale distortions (e.g., GW with horizontal wavelengths larger than the observation volume) and ripples or wave-like features.

## 4 Results

### 4.1 Kelvin-Helmholtz instabilities

In June 2013, MAARSY was used to conduct a multi-beam experiment campaign. During this period, the PMSE strength was rather variable regarding the duration and its horizontal and vertical extension. On the 21 June, we observed an interesting PMSE structure with several thin layers showing signs in the morphology, which seem to evolve into KHI. Figure 6 shows the





SNR, the radial velocity and the spectral width of the vertical beam. There are times when the morphology of the PMSE layer is forced by strong upward and downward motions, which appear in the radial velocity as well. We identify two possible KHI events around 00:30 UTC and around 15:45 UTC.

An advantage of the radar measurements is the Doppler information from which we obtain the 3D winds independent of the cloud conditions and during daylight. In particular, we are able to investigate the stability of the flow when the KHIs occur. MR data is used to extend the altitude coverage and to complement the MAARSY winds. Figure 6 shows both observations. We show both horizontal wind components obtained from MAARSY in panel a) and b). Panel c) and d) present the meteor radar zonal and meridional winds. The zonal and meridional winds are dominated by the tides. Further, the zonal wind reverses at approximately 88 km, separating the westerly mesospheric jet in the lower part from the easterly thermospheric jet above. The wind reversal and the strong semi-diurnal tide generate pronounced wind shears in the flow. Differences in the wind magnitude between both observations are mainly attributed to the different sampling volumes (80 km diameter for MAARSY and 400 km diameter in the case of the MR) and the temporal resolution (approx. 4 minutes for MAARSY and 30 Minutes for the MR winds). However, the MR as well as the MAARSY winds reflect the large scale features as the diurnal tidal pattern and the strong zonal pattern in the altitude range from 80-90 km.

It is known that KHI evolve in dynamical unstable flows due to strong shears with $Ri = N^2/S^2 < 0.25$ (Miles and Howard, 1964). Where $N$ is the Brunt-Väisälä frequency and $S$ describes the horizontal wind shear. The Brunt-Väisälä frequency is computed from:

$$N = \sqrt{\frac{g}{T}\left(\frac{dT}{dz} + \frac{g}{c_p}\right)} \quad . \tag{1}$$

Under the assumption of a mesospheric temperature close to the mesopause of $T = 130$ K, a temperature gradient of $\frac{dT}{dz} = 0$, a gravitational acceleration of $g = 9.64 \; m/s^2$, and a specific heat at constant pressure of $c_p = 1009 \; J/kgK$, we obtain a Brunt-Väisälä period of approximately 4 min at the altitude of the PMSE. We also estimated the Brunt-Väisälä period above and below the PMSE layer using a NRLMSIS00 profile (Picone et al., 2002).

The obtained $Ri$ are shown in Figure 6 assuming a NRLMSIS-00 background temperature profile shifted to match 130 K at the mesopause. The upper panel visualizes the Ri obtained from MAARSY and the lower panels show the MR derived results. In particular, the MAARSY data shows that there are often low $Ri$ within the PMSE layer, which is expected considering that turbulence is an important factor in the formation of PMSE (Rapp and Lübken, 2004). Similarly, $Ri$ are obtained from the MR, although the coarser vertical resolution as well as the vertical averaging in the wind analysis has to be taken into account to estimate the $Ri$. Our wind measurements confirm that KHI occurred during times showing a strong vertical wind shear in the horizontal wind speeds that could have generated a sufficiently small $Ri < 0.25$ supporting the notion that the instabilities are of a dynamical origin. However, as we use an empirical temperature background profile, the low $Ri$ values indicate in first place a strong vertical wind shear instead of absolute measurements of $Ri$.

The wave characteristics are derived applying a Stokes analysis (Vincent and Fritts, 1987; Lue et al., 2013). In a first step, we computed the wavelet spectra for all three components (Torrence and Compo, 1998). Figure 6 shows the resulting spectra for the zonal and meridional wind component after the subtraction of the mean wind and the tidal components. The two panels



indicate two wave bursts with periods $T < 30$ min, which coincide with the occurrence times labeled by the red boxes in Figure 6. For the first KHI event, we observed a mean period of $11.5 \pm 1.5$ min in all three wind components (zonal, meridional and vertical) between 00:00-00:50 UTC. The second KHI occurred between 15:00-16:00 UTC and had a mean observed period of $20.3 \pm 1.0$ min in all three wind components. However, the wavelet spectra shown in Figure 6 also indicates some spread in

the observed periods suggesting some dispersion between the events.

The mean horizontal wavelength of the first group of KHI was determined to be $\lambda_z = 10.7 \pm 5$ km and for the second event we estimated a wavelength of $\lambda_h = 12.3 \pm 5.3$ km. These values are obtained assuming that the KHI are advected by the mean winds through the radar beam. The vertical extension of the KH billows is estimated from our RTI to be $\lambda_z = 3 \pm 0.5 km$. Fritts et al. (2014) conducted DNS simulations to characterize the KHI evolution at the MLT and investigated the evolution of

KHI in dependence of the mean shear flows and GW induced shear flows for small $Ri = 0.05 - 0.20$. The DNS simulations are compared to actual NLC observations (Baumgarten and Fritts, 2014). This leads to a depth-to-wavelength ratio of 0.3-0.4 suggesting a small initial Ri (Werne and Fritts, 1999, 2001).

In Figure 6, we zoom in on the SNR, the vertical velocity and the spectral width for both KHI. The SNR indicates a train of ripples passing through the vertical beam. Such structures are rather common in airglow images (Hecht et al., 2005, 2007).

Depending on the temporal evolution of the KHI, we observe strong vertical motions as visualized in Figure 6 c) and d). Considering PMSE as an inert tracer, the layer follows the upward and downward motion of the propagating billows. After the passage of the KHIs, the layer appears to be smoother and vertically smeared compared to the more confined structure that existed before, which is likely related to the turbulence generated by the KHIs. From our spectral width measurement in Figure 6 panel e) and f), we obtained that the vertical motion of the KHIs is accompanied by an increased spectral width, which

is associated with an increased turbulence generation.

We further identified the presence of some gravity waves that become unstable and likely generated the ripple structures. In Figure 6 we present the zonal and meridional wavelet spectra for three altitudes at 83 km, 90 km and 95 km. The wavelet spectra show the MR wind after removing the tides and mean flow. The pictures points out that there are some components with GW-like periods with significant amplitudes at 83 km altitude, which more or less disappear at 90 km and than grow

again. This is in particular obvious for the zonal wind component and to a smaller degree for the meridional wind.

## 4.2   Gravity Wave statistics using horizontally resolved radial velocities

In previous section, we presented results of ripples/billows causing modulations in rather thin PMSE layers. However, there are times where PMSE covers a much larger vertical and horizontal volume. Sometimes the complete scanning area shown in

Figure 1 was filled with PMSE and provided a sufficient strong backscatter signal to get a reliable radial velocity measurements for each beam direction. This permits to construct radial velocity maps of the horizontal wind variability caused by ripples or GWs propagating through the layer.

Horizontally resolved radial velocities maps using PMSE as passive tracer for neutral dynamics were already introduced by Stober et al. (2013). Here, we apply this method to enhance our statistics, investigating several days of our multi-beam



observations from 21 June to 30 June 2013. Considering our previous experience retrieving monochromatic GW properties from horizontally resolved radial velocity images, we modified the experiment to ensure that our sampling time (time for a complete scan) is faster than the Brunt-Väisälä period for the summer mesopause, which is around 4 minutes. Further, we improved our analysis to fit directly for the horizontal wavelength, propagation direction and phase speed.

Before we can extract ripple or GW features from our radial velocity measurements, we need to remove the mean wind and large scale distortions or contributions from waves with scales larger than our observation volume. Therefore, we fit for the wind field using the VVP approach (Browning and Wexler, 1968; Waldteufel and Corbin, 1979). The basic idea is to drop the assumption that the wind field has to be homogenous within the observation volume. Browning and Wexler (1968) expressed the wind field by a Taylor series,

$$
\begin{aligned}
u &= u_0 + \frac{\partial u}{\partial x}(x - x_0) + \frac{\partial u}{\partial y}(y - y_0) \\
v &= v_0 + \frac{\partial v}{\partial x}(x - x_0) + \frac{\partial v}{\partial y}(y - y_0) \ .
\end{aligned}
\tag{2}
$$

Here, $u_0$ and $v_0$ express the mean zonal and meridional wind in the observed volume, respectively, and $\partial u/\partial x$, $\partial u/\partial y$ and $\partial v/\partial x$, $\partial v/\partial y$ a zonal and meridional wind gradient in $x$ and $y$ direction. For simplicity, we assume that the radar is located at $x_0 = 0$ and $y_0 = 0$. Although the first order approach outlined here does not account for all the variability within the observation volume, it provides a good approximation of the mesoscale situation. The first order zonal and meridional gradient terms in x- and y-direction can be associated to waves and inhomogeneities larger than the observation volume.

We use the first order wind approximation given above to retrieve smaller structures within our field of view. We decompose each radial velocity measurement by subtracting the VVP solution to obtain a radial velocity residual $v_r^{res}$ for each beam,

$$
v_r^{res} = v_r^{obs} - v_r^{VVP} \ .
\tag{3}
$$

Here, $v_r^{obs}$ is the individually observed radial velocity for each beam. In fact, the radial velocity residuum includes now the wind variability smaller than the observation volume.

A sequence of 9 successive radial velocity images containing all different beam directions is shown in Figure 6. The measu-
rements were taken on 30 June 2013 and are representative for the type of features that can be seen in these images. Some frames show rather coherent and wave-like features, other images seem to be more dominated by random structures, which may be caused by the superposition of several waves/ripples moving through our field of view. Whenever we found a coherent wave-like structure in these images that lasted at least three successive frames, we tried to fit the wave features, i.e., horizontal wavelength, phase speed and propagation direction.

Following Fritts and Alexander (2003), a gravity wave is described in the linear theory by:

$$
(u, v, w) = (u', v', w') \cdot e^{i(kx + ly + mz - \omega t) + \frac{z}{2H}} \ .
\tag{4}
$$

Here, $u'$, $v'$ and $w'$ are the zonal, meridional and vertical amplitude of the GW respectively, $k$ and $l$ are the zonal and meridional horizontal wave numbers, $m$ denotes the vertical wave number, $\omega$ is the Eulerian GW frequency and $H$ is the scale height, respectively. As we just observe the horizontal structure of the wave, we modify Eq. 4 by introducing a phase $\varphi = mz - \omega t$.





Further, we can neglect the term for the amplitude growth with altitude, $\frac{z}{2H}$, as we only have information about the wave at a fixed altitude. Thus, we can rewrite Eq. 4,

$$(u, v, w) = (u', v', w') \cdot e^{i(kx+ly+\varphi)} \ .$$ (5)

Assuming only a slow change of the intrinsic frequency and vertical wavelength over successive frames, we infer the vertical wave number $m$ and the frequency $\omega$ using the time derivative of the phase $\varphi$,

$$\frac{d\varphi}{dt} = -\omega \ .$$ (6)

The advantage of the outlined procedure is that we directly obtain the intrinsic wave/ripple characteristics. The intrinsic wave frequency $\hat{\omega}$ is straight forward computed as we know the horizontal wavelength, the propagation direction and the mean mesoscale wind components.

In total, we were able to identify 15 ripple events within the 9-day campaign. For that, we searched all 2D radial velocity images for coherent wave like features that lasted at least three successive frames. The obtained parameters of these ripples are
presented in Figure 6 with regard to the (intrinsic) period, the (intrinsic) phase speed, horizontal and vertical wavelength. The duration of the GW/ripples within the scanning volume varied between 10-50 minutes. Most of the events lasted approximately 20-25 minutes, viz. for more than 4 frames. We did not obtain intrinsic periods shorter than the Brunt-Väisälä period, but they are already rather close to this limit. Remarkable is that the computed periods are much larger with 20-90 minutes, as most of these waves/ripples move against the background mesoscale flow. This behavior appears also in the intrinsic and observed
phase speeds. Intrinsic phase speeds had values between 50-90 m/s, whereas the observed phase speeds have values between 2-23 m/s. At phase speeds faster than 60 m/s the wave.like structures would travel more than the diameter of the radar beam, and, hence the positive and negative phase fronts would cancel each other. Due to the size of the scanning volume and the subtraction of the mesoscale variability, the horizontal wavelengths represent the characteristic scale of our scanning volume between 20-40 km. The obtained vertical wavelengths are rather short and with values ranging from 5 to 10 km.
The polar diagram in Figure 11 shows the distribution of the propagation direction for all 15 events. For simplicity, we just indicated the mean wind direction with a red arrow. However, individual ripples moved at a certain angle to the prevailing winds. These angles between the prevailing wind and the propagation direction of the wave/ripple showed values between 90-180°.

## 5  Discussion

Analyzing the wind structure from MAARSY as well as the MR wind observations, we showed that the observed modulations in the morphology of PMSE are likely caused by dynamical instability. This is supported by the computed Richardson number $Ri < 0.25$, which is related to a strong shear flow. Further, we are able to show that there are mesoscale GW present during the observation of the KHI. These mesoscale GW show significant amplitudes at the height of the KHI, a much smaller amplitude 5 km above and an increased amplitude at 95 km altitude. According to Lindzen (1988), upward propagating waves do not



dissipate their whole energy at once. They dissipate some energy decreasing the GW amplitude and start growing again above, which is well represented in the MR data.

Other observations above Andenes suggest that KHI seem to occur frequently at the polar MLT. Fritts et al. (2014) and Baumgarten and Fritts (2014) analyzed two KHI events from a NLC camera network and inferred the background wind si-

tuation by tracing the ice clouds over a much larger field of view than the billows. They retrieved a horizontal wavelength of 20-30 km for the breaking/generating GW. Baumgarten and Fritts (2014) investigated different events and derived horizontal scales of 5-10 km for the KHI, which are in remarkable agreement to our measurements of 7-12 km. Due to the weather conditions there is no data available to perform a direct comparison with our data. Comparable horizonal and vertical wavelengths are also reported from mid-latitude GRIPS (GRound based Infrared P-branch Spectrometer) measurements (Wüst et al., 2017).

Comparing our results to previous optical observations from a NLC camera at Trondheim monitoring the MLT above Andenes suggest slightly higher observed phase speeds and much shorter observed wave periods (Demissie et al., 2014). The horizontal wavelengths are comparable and had values between 20-40 km. However, these measurements were taken between summer 2007 and summer 2011 and do not cover the period presented in this manuscript.

Our observations are also consistent to what was reported previously from airglow observations, although these measurements

were not taken at the same location. Hecht et al. (2005) showed that breaking mesoscale GW form ripple structures that evolve into KHI. Hecht et al. (2007) derived some statistics of ripples and investigated whether they were of dynamical or convective origin. They inferred the atmospheric stability from lidar and MR winds. The described features are similar to what we observed with MAARSY.

Comparing our statistical results with the model data from Horinouchi et al. (2002) provides further indication that we have

observed dynamically unstable KHI. Their model resolved convectively generated mesoscale gravity waves that propagate up to the mesosphere. Once the GW reach the mesosphere they become dynamically instable and form the ripple features. They obtained a typical billow scale of 8-15 km that occupy a region with 30-50 km in diameter. The model shows that such events last about 25-40 minutes. Considering our observations, we likely observed similar scales and durations, which reassure us that most of the observed events are dynamically instable KHI. However, we cannot absolutely rule out the possibility that some of

the wave-like features were of convective origin. Such secondary instabilities can occur after KHIs (Hecht et al., 2005).

Further we have to point out that in the model from Horinouchi et al. (2002) the source of the mesoscale GW are convective clouds. This is likely not the case at Andenes. We assume that tropospheric jet instabilities are the more likely source of mesoscale GW at polar latitudes. A detailed discussion about the wave sources is beyond the scope of this paper and requires dedicated model runs.

# 6   Conclusions

In this study we present unique observations of ripples and KHI in the wind field during full daylight conditions at polar latitudes above Andenes. The observation were conducted with MAARSY during summer 2013 using PMSE as tracer for neutral





dynamics. The wind anaylsis was complemented with data from a collocated MR in order to infer and validate the mesoscale GW activity.

We were able to identify two KHI from the morphology of the PMSE layer and estimated the characteristic scale of this billows to be in the order of 8-12 km. Our measurements indicate an increased spectral width accompanied with an increased

turbulence while the billows occurred in the vertical beam. In addition, we inferred from the MR observations the presence of mesoscale GW that dissipated a part of their energy between 83 and 90 km altitude and an amplitude growth above, which is at least in qualitative agreement to Lindzen (1988).

Further, we demonstrated that multi-beam experiments are suitable to directly obtain ripple properties as horizontal wavelength, intrinsic frequency and propagation direction as well as the propagation direction. The observed values are in reasonable

agreement to model simulations of breaking mesoscale GW generated from convective tropospheric clouds (Horinouchi et al., 2002). and also consistent to airglow observations (Hecht et al., 2005, 2007).

*Data availability.* The data is available upon request by stober@iap-kborn.de.

*Competing interests.* There are no competing interests.

*Acknowledgements.* We thank the IAP technical staff for keeping MAARSY operational. The helpful discussions about gravity waves with Peter Hoffmann are acknowledged. This work was partially supported by the WATILA Project (SAW-2015-IAP-1). S. Sommer was funded by ILWAO (SAW-2012-IAP-4). MAARSY was built under grant 01LP0802A of the German Federal Ministry of Education and Research. Some of the contributing researchers are supported by the German research grant MATMELT (SAW-2014-IAP-1). C. Schult was supported by grant STO 1053/1-1 (AHEAD) of the Deutsche Forschungsgemeinschaft (DFG).



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





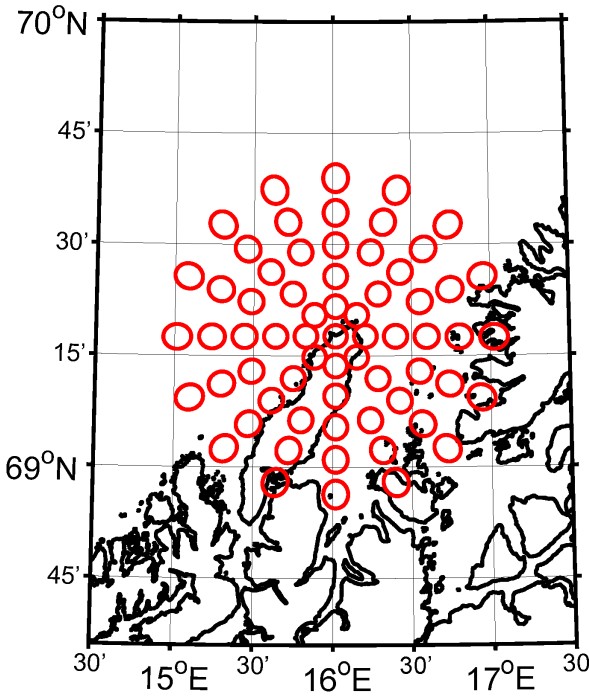

**Figure 1.** Projection of MAARSY beam positions for the multi-beam experiments during summer 2013. The red circles show the illuminated radar beam area assuming a beam width of 3.6° at 84 km altitude. The black lines are the shore line around the north Norwegian island Andøya.

**Table 1.** Experiment Parameters (PRF: Pulse Repetition Frequency, CI: Coherent Integrations, acq: acquisition, Pulse code: coco - 16 bit complementary code)

| Parameter | Exp 1 | Exp 2 | Exp 3 | Exp 4 |
|---|---|---|---|---|
| PRF (Hz) | 1250 | 1250 | 1250 | 1250 |
| sampling resolution (m) | 300 | 300 | 300 | 300 |
| CI | 2 | 2 | 2 | 2 |
| Pulse code | coco | coco | coco | coco |
| number of beams | 17 | 17 | 17 | 17 |
| off-zenith angles | 0,5,10° | 0,15° | 0,20° | 0,25° |
| Nyquist velocity (m/s) | 22.5 | 22.5 | 22.5 | 22.5 |
| data points | 256 | 256 | 256 | 256 |
| acq. time (seconds) | 25 | 25 | 25 | 25 |



**Figure 2.** Stack plot of the radial velocity measurements with MAARSY. The lower panels shows the color coded statistical uncertainties of the radial velocity measurement.





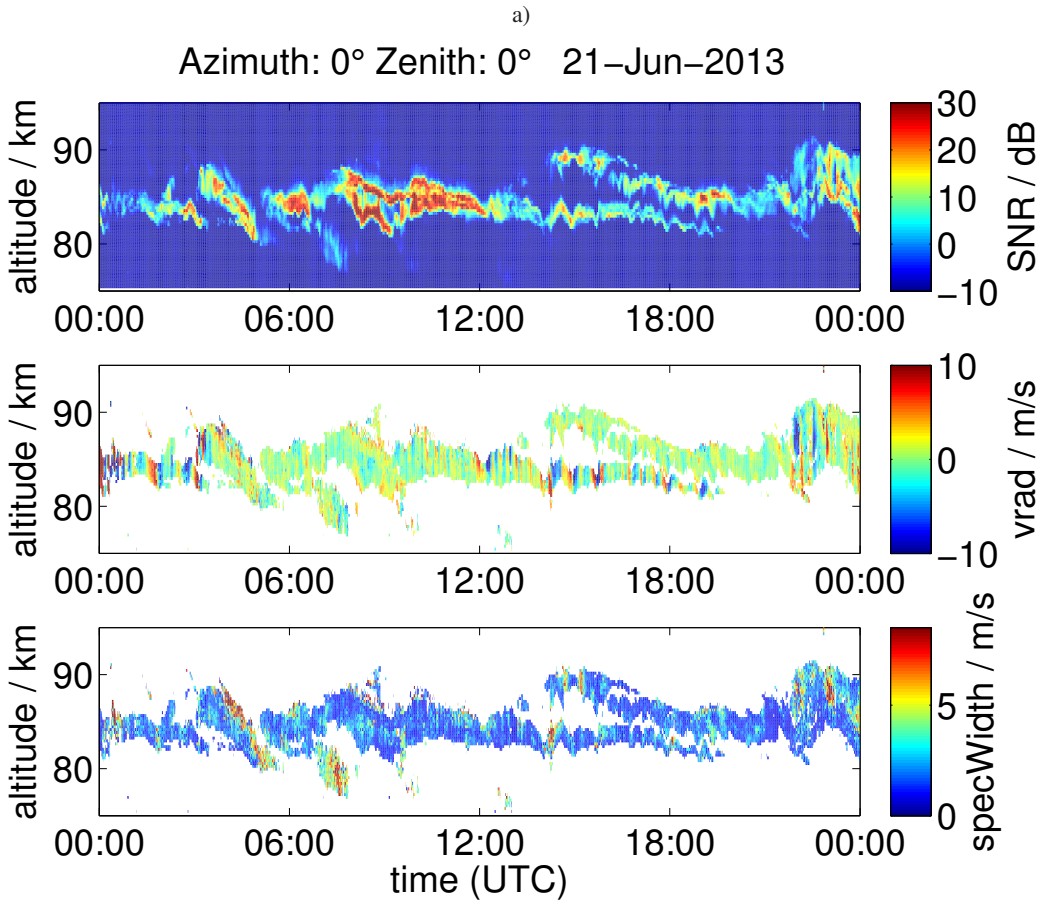

**Figure 3.** Measured SNR from the vertical beam for 21. June 2013. Radial velocity determined from spectral analysis using a truncated Gaussian fit. Computed spectral width for the vertical beam.





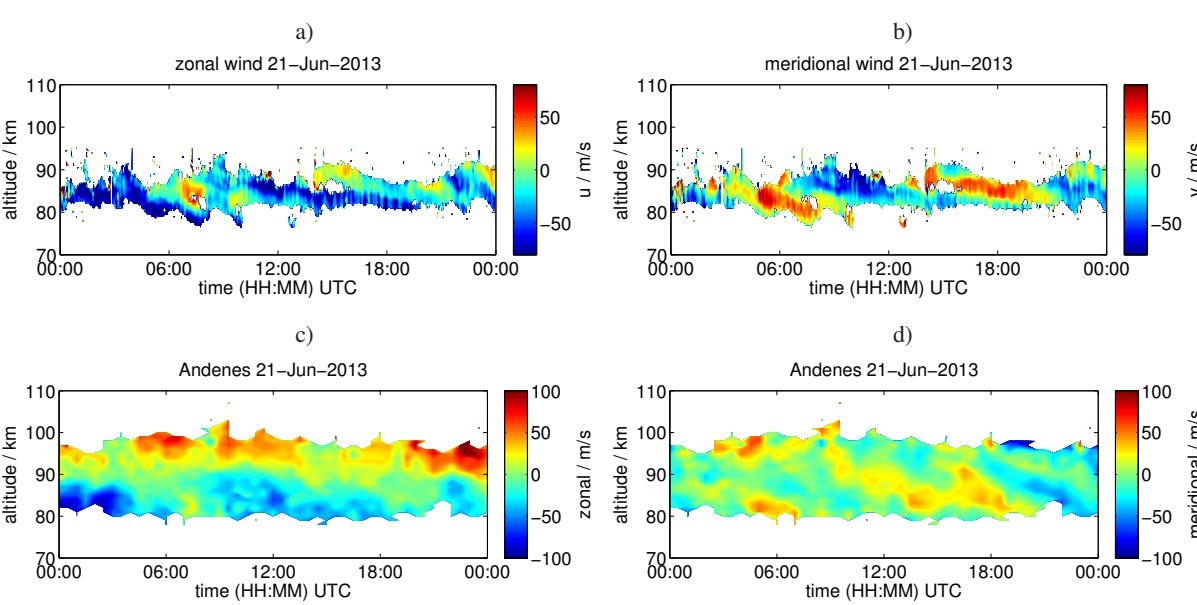

**Figure 4.** Color coded zonal and meridional winds for 21 June 2013. Panel a) and b) show the observations from MAARSY. The lower panels c) and d) show the data from the collocated meteor radar.





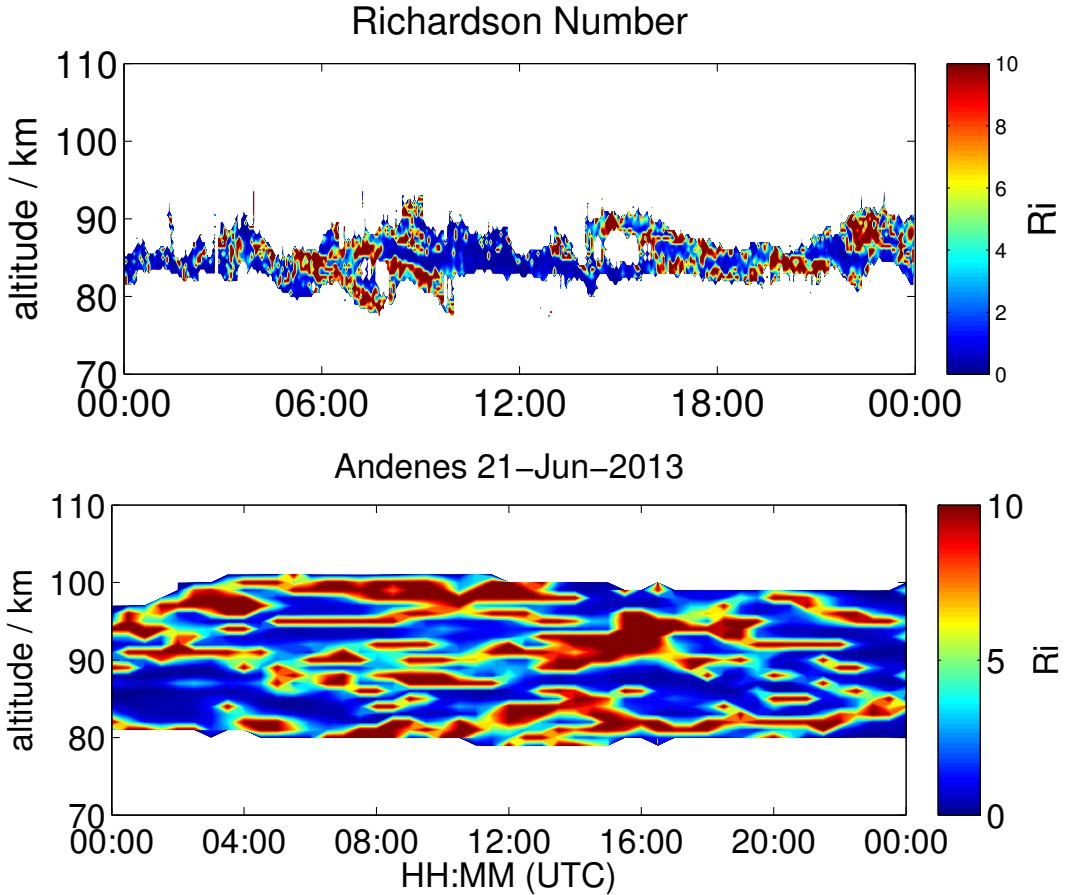

**Figure 5.** Richardson number estimated from the vertical wind shear and Brunt-Vaïsälä frequency.

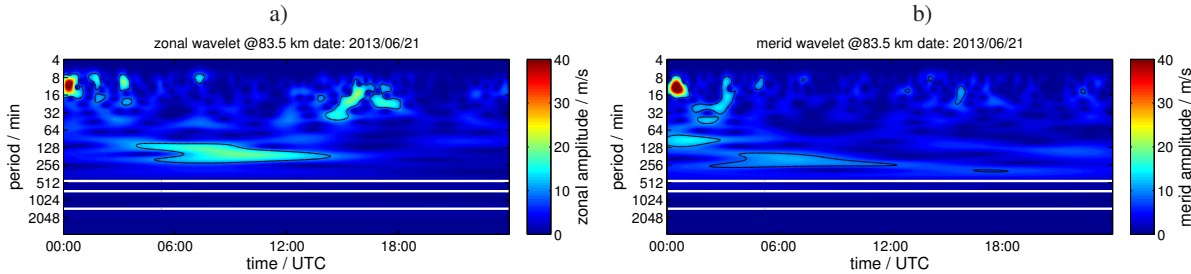

**Figure 6.** Wavelet spectra of the zonal and meridional wind using the MAARSY wind measurements after the diurnal, semi-diurnal and terdiurnal tide was removed.





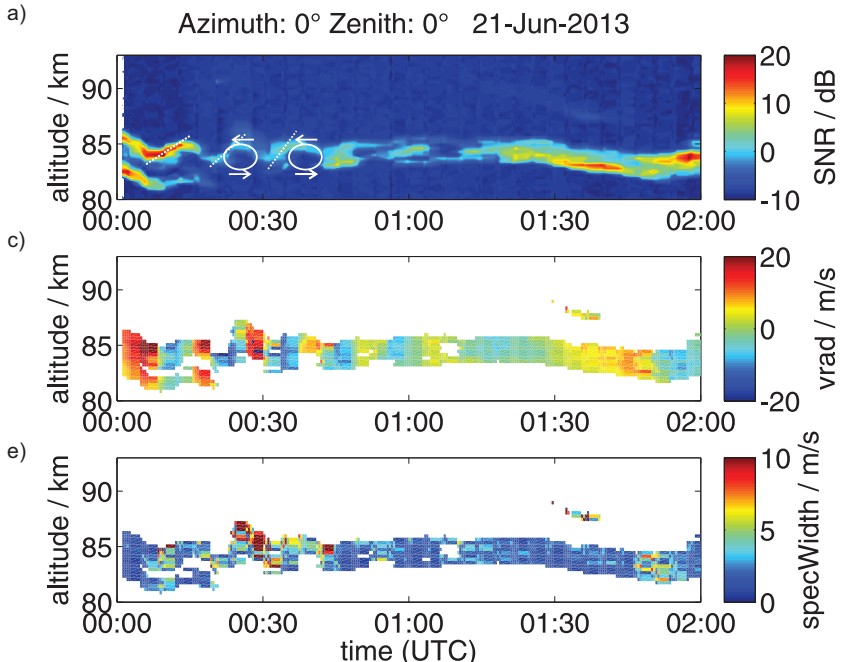

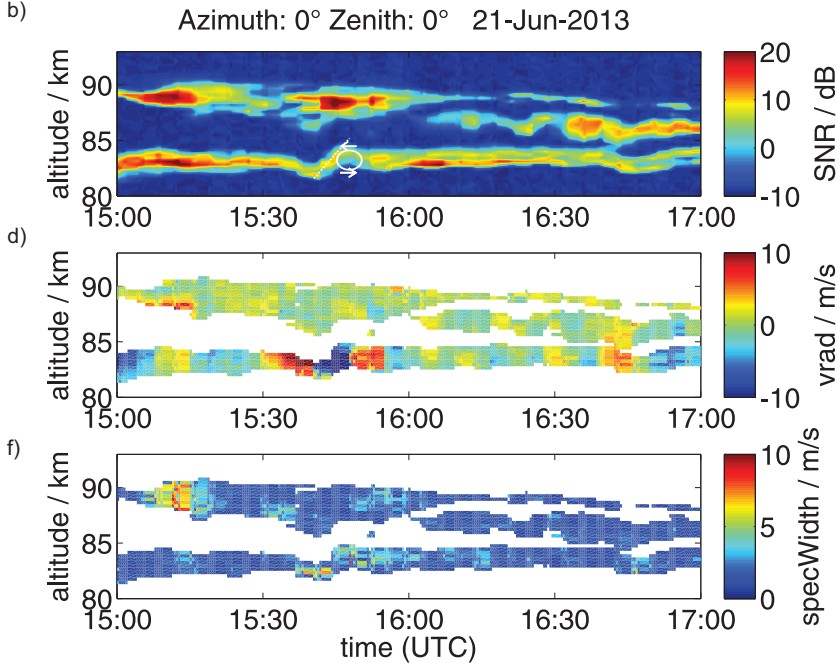

**Figure 7.** Zoom in on the SNR (a) and (b), radial velocity (c) and (d) and spectral width (e) and (f) for the two Kelvin Helmholtz Instabilities observed on 21 June 2013.





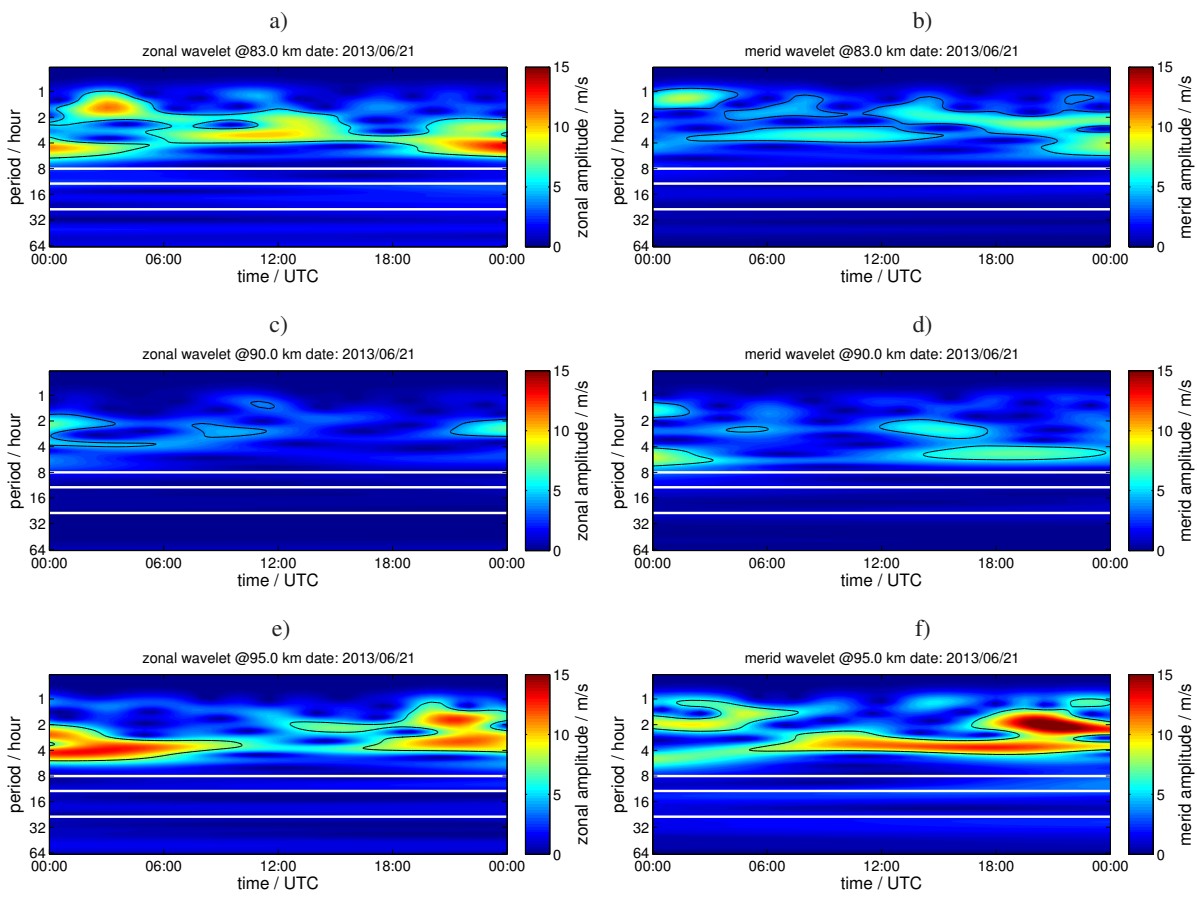

**Figure 8.** a)-f) Zonal and meridional amplitude wavelet spectra for different altitudes. The time series were filtered to remove the mean wind as well as the tidal components. The white lines indicate the periods of the diurnal, semi-diurnal and terdiurnal tide.



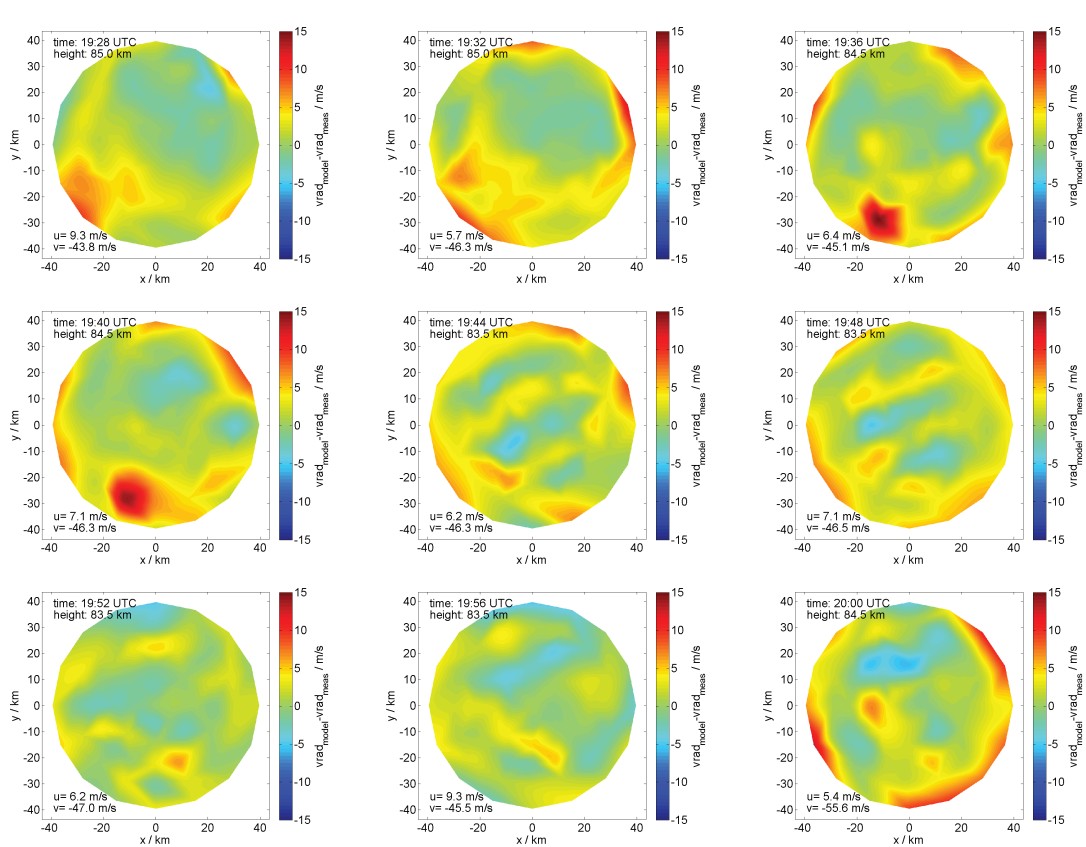

**Figure 9.** Sequence of 9 successive radial velocity residual images measured on 30 June 2013.





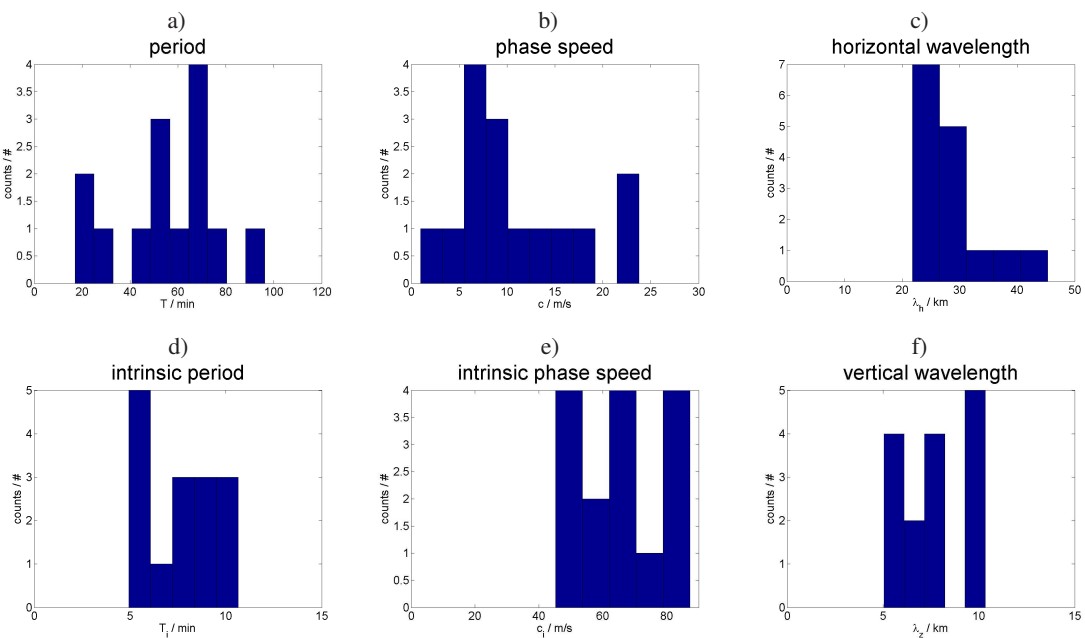

**Figure 10.** a) Histogram of the observed GW periods. b) Histogram of the determined phase speeds using the 2D fit. c) Obtained horizontal wave lengths. d) Histogram of derived intrinsic GW periods. e) Statistics of the computed intrinsic phase speeds. f) Histogram of estimated vertical wave lengths assuming linear theory.



**Figure 11.** Polar diagram of GW/ripple propagation direction. The red arrow denotes the mean wind direction.