# Peer review of "Observation of Kelvin-Helmholtz Instabilities and gravity waves in the summer mesopause above Andenes in Northern Norway"

_Atmospheric Chemistry and Physics, 2017_

## Referee Comment (RC1) · Anonymous Referee #1 · 16 Jan 2018

General Comments

The paper presents new PMSE data obtained with MAARSY radar in Northern Norway and simultaneous meteor radar winds. The data suggest the presence of two brief Kelvin-Helmholtz (KH) events on one particular day. The authors also present the analysis of 15 short-period gravity wave events based on volume velocity processing (VVP) of the multi-beam Doppler radar data.

The manuscript is very difficult to review since the authors did not process the figure numbers correctly. Almost all figures are called Figure 6.

The KH billows are not easy to see, presumably due to the relatively low height and

time resolution. They are inferred from the observed aspect ratio and wind shear. The authors quote earlier MST observations in the troposphere and mesosphere. It should be included that the SOUSY radar observed KH events in the mesosphere three decades ago (Reid et al, Nature, 1987).

The VVP method has been described already in 2013. Nevertheless, the identification of short period and short wavelength gravity waves is an achievement with the MST radar.

The discussion is kept general, since no other simultaneous measurements are available. There are many other observations of KH in the mesosphere (e.g. airglow, NLC), so it is not a new discovery. It is, however, the first report of KH from the MAARSY radar.

The comparison of the observed wave characteristics with the model results from Horinouchi et al (2002) seems only of limited usefulness, since they describe very different dynamical conditions.

I recommend to accept with minor revisions.

Specific Comments

p. 3. Praise: It is appreciated to include detailed description of the outlier treatment.

p. 5 l. 23. Why not use a local climatology of temperature (Lüben, Falling Spheres?)

p. 8 l. 5. For completeness, it would be good to include the relation for intrinsic and Doppler shifted quantities.

p. 8. l. 29. I doubt that an old theoretical result (Lindzen 1988) can do justice to interpreting the observations compared to more modern simulations of GW breaking (e..g. Fritts, Hickey, Snively). The authors should try to compare with these more comprehensive models.

p. 8 l. 5. For completeness, it would be good to include the relation for intrinsic and

Doppler shifted quantities.

Technical Corrections

AGAIN: All Figure references must be fixed!

p. 6. l. 1. no red boxes

p. 6 l. 6: lambda_h = 10.7 km (horizontal)

Fig. 10. Better use same y-scale for counts.

---

## Author Comment (AC1) · 26 Jan 2018

The authors apologize for the latex mistake that lead to an incomplete labelling of all figures and tables in the manuscript.

We uploaded a corrected version of the pdf as supplement material and kindly ask the readers to use this updated mauscript.

Please also note the supplement to this comment: https://www.atmos-chem-phys-discuss.net/acp-2017-1170/acp-2017-1170-AC1-supplement.pdf

[Figure]

[Figure]

**Supplement:**

[revised manuscript text omitted]

---

## Referee Comment (RC2) · Anonymous Referee #2 · 1 Feb 2018

The manuscript reports MST radar observations of KHI based on the structural evolution of PMSE. These are the first MST radar observations of the kind and thus, well worth publishing. The authors further provide a set of basic characteristics based on 15 wave events which are then compared to earlier observations of similar activity captured with different measurement techniques. Much of the information is well described in the manuscript but some further clarifications are necessary before publishing. Most of the questions below are asking for more explanations and discussion of the key features in the data plots.

[Figure]

**Main comments:**

- PMSE is a major side topic of the paper because it is used as a tracer. The manuscript would benefit from a short characterization / climatology section of PMSEs in the introduction to let the readers know what are the normal PMSE conditions, structure, thickness, lifetime, occurrence etc., so what are the KHI observations in this study compared to. The references to earlier work are given but their main results should be outlined.

- Figure 1 is said to show all beam positions as well as the ones for each experiment. However, the figure shows map projections of all beams. How were the 17 beams for each experiment chosen and what were they?

- A meteor removal procedure is explained in section 3. Is that a standard method with an existing reference to earlier work, or is it implemented here to improve the current analysis? In the latter case the thresholding would benefit from some justifications.

- Meteor radar data are used in addition to MAARSY data but the data description for meteor radar is very thin compared to that of MAARSY. In particular, a height range and resolution for meteor radar data should be included in the corresponding paragraph on page 4.

- When the first identification of KHI is presented (page 5, line 3), a short description of what the reader is supposed to look at in the figure would help a lot. What are the changes in the different parameters which give away the KHI occurrence? In the same paragraph earlier the meaning of: "several thin layers showing signs in the morphology" is unclear. Do you mean that the PMSE structure consists of several transient thin layers?

- The description of Figure 5 says that there are often low Ri values within the PMSE. Not sure what is low and high, but it is hard to see if there really is more high or low values in the two colour panel. Is there a way to justify that statement?

- What are the GW-like periods you focus on in Figure 8? It seems like there is significantly less wave activity during the latter KHI event?

- Line 8 on page 5 says:"The zonal and meridional winds are dominated by the tides." A sentence saying how the data plot supports this idea would be helpful.

- When Richardson number is being introduced in section 4.1, a brief reasoning for why that is a useful parameter in the KHI study would be good.

- Can you specify the meaning of "coherent wave-like structure" (page 7, line 23)? What is coherent enough?

- What is the significance or implication of the prevailing wind direction with respect to the ripple propagation direction? The observations and the plot are not really discussed.

- When introducing earlier observations by Demissie et al. (page 9, line 12), you mention that those are from different years. Does that mean that you would expect annual differences?

**Minor comments:**

- The introduction mentions "mesoscale" many times. It is a relative measure which depends on the observations, so it would be good to give a rough number or range for it.

- The paragraph change on page 1 and line 23–24 is unnecessary when the first sentence of the new paragraph refers to the last sentence of the previous paragraph.

- On page 2 and line 9, "and many other" is redundant since the reference list starts with "e.g.".

- On page 3 and line 21, Figure 2 is hardly a contour plot.

- On page 4 and line 4, should the "zenith distance" be a zenith angle since the rest of the sentence talks about degrees?

- Figured 3 has an "a)" as a panel marking but I do not see the panel labels b and c.

- The description of Figure 5 in the text says that there is Ri calculated from MAARSY and MR data (the plural s in "lower panels" seems redundant). The figure caption says that the panels are Ri from wind shear and Brunt-Väisälä frequency. Could you add the data source information in the figure caption to make it more self-explanatory?

- Figure 3 does not show any red boxes but based on the statement on the top of page 6, it might make a difference to generate the boxes.

- What is DNS simulation (page 6, line 9)?

- Does "rather common" (page 6, line 14) mean "not uncommon", or is there actually a description attached to it, which could be added to be a bit more precise?

- On page 6 and line 24: "than" should probably be "then"

- "Train of ripples" referring to Figure 7a seems right, but in Figure 7b it looks more like one single wave-like feature.

- Should be "wave-like" instead of "wave.like" on page 8 and line 16.

- On page 9 and line 8 the text blames weather conditions for not having other observations. Does that relate to a lack of optical observations due to the daylight conditions, or does it really mean weather?

- Seems like there is an extra "propagation direction" on page 10 and line 9.

---

## Author Comment (AC2) · 12 Apr 2018

General Reply to Reviewer #1: The authors thank the reviewer for taking his time to provide the comments below. We are grateful for the additional recommended publications. We also want to apologize for the mistake in the figure labelling. The comments raised are answered comment by comment. Changes in the manuscript are highlighted by latexdiff.

Comment: The authors quote earlier MST observations in the troposphere and mesosphere. It should be included that the SOUSY radar observed KH events in the mesosphere three decades ago (Reid et al, Nature, 1987). Reply: We thank the reviewer for

pointing at the publication of Reid et al. 1987.

Comment: The VVP method has been described already in 2013. Nevertheless, the identification of short period and short wavelength gravity waves is an achievement with the MST radar. Reply: Compared to the publication in 2013 we improved the previous VVP methodology by extracting the horizontal information by a non-linear fitting routine to increase the degree of automatization.

Comment: The discussion is kept general, since no other simultaneous measurements are available. There are many other observations of KH in the mesosphere (e.g. airglow, NLC), so it is not a new discovery. It is, however, the first report of KH from the MAARSY radar. Reply: Observations of KHI in the MLT are reported from different observational techniques. The analysis presented here on MAARSY observations are motivated to show that such small scale structures can be identified in the radial Doppler measurements using PMSE as tracer.

Comment: The comparison of the observed wave characteristics with the model results from Horinouchi et al (2002) seems only of limited usefulness, since they describe very different dynamical conditions. Reply: The simulations presented in Horinouchi are useful to bring a more general understanding on how instabilities can evolve for different sources. However, we agree that the dynamics describe in Horinouchi is different to what we can observe at polar latitudes.

Specific Comments

Comment: p. 3. Praise: It is appreciated to include detailed description of the outlier treatment. Reply: We modify this paragraph to make it better understandable on how meteors are removed. However, the presented numbers and thresholds depend on how the experiment is analyzed and also on the specifics of the radar. Other systems may require different removal strategies.

Comment: p. 5 l. 23. Why not use a local climatology of temperature (Lüben, Falling

Spheres?) Reply: As shown in Luebken et al., 1999 there is a fairly acceptable agreement between the falling spheres and the MSIS-model.

Comment: p. 8. l. 29. I doubt that an old theoretical result (Lindzen 1988) can do justice to interpreting the observations compared to more modern simulations of GW breaking (e..g. Fritts, Hickey, Snively). The authors should try to compare with these more comprehensive models. Reply: The recommended papers are now discussed in the manuscript. The observed rather high phase speeds of the wave-like structures indeed indicates likely wave ducting as discussed in Snively.

Comment: p. 8 l. 5. For completeness, it would be good to include the relation for intrinsic and Doppler shifted quantities. Reply: We added both equations to the manuscript.

Comment: Technical Corrections AGAIN: All Figure references must be fixed! p. 6. l. 1. no red boxes p. 6 l. 6: lambda_h = 10.7 km (horizontal) Fig. 10. Better use same y-scale for counts. Reply: Done.

---

## Author Comment (AC3) · 12 Apr 2018

General Reply to Reviewer #2: The authors thank the reviewer for taking his time to provide the comments below. We also want to apologize for the mistake in the figure labelling. The comments raised are answered comment by comment. Changes in the manuscript are highlighted by latexdiff.

Comment: PMSE is a major side topic of the paper because it is used as a tracer. The manuscript would benefit from a short characterization / climatology section of PMSEs in the introduction to let the readers know what are the normal PMSE conditions, structure, thickness, lifetime, occurrence etc., so what are the KHI observations in this study

compared to. The references to earlier work are given but their main results should be outlined. Reply: We added a few sentences in the introduction to describe PMSE in general and in particular related to MAARSY measurements.

Comment: Figure 1 is said to show all beam positions as well as the ones for each experiment. However, the figure shows map projections of all beams. How were the 17 beams for each experiment chosen and what were they? Reply: The manuscript is now more precise about the experiment description. A detailed list of the differences between each beam sequence is given in Table 1.

Comment: A meteor removal procedure is explained in section 3. Is that a standard method with an existing reference to earlier work, or is it implemented here to improve the current analysis? In the latter case the thresholding would benefit from some justifications. Reply: The meteor removal was done and is likely non-standard. The mentioned thresholds and numbers depend on the experiment settings and how the data is analyzed. Different radars or experiments may require some tuning of the filtering values to ensure a similar quality.

Comment: Meteor radar data are used in addition to MAARSY data but the data description for meteor radar is very thin compared to that of MAARSY. In particular, a height range and resolution for meteor radar data should be included in the corresponding paragraph on page 4. Reply: We added a reference and a sentence describing the meteor radar winds and the vertical and temporal resolution. To ensure consistency we updated the wind pictures.

Comment: When the first identification of KHI is presented (page 5, line 3), a short description of what the reader is supposed to look at in the figure would help a lot. What are the changes in the different parameters which give away the KHI occurrence? In the same paragraph earlier the meaning of: "several thin layers showing signs in the morphology" is unclear. Do you mean that the PMSE structure consists of several transient thin layers? Reply: We added a few sentences what we believe are the KHI

signatures.

Comment: The description of Figure 5 says that there are often low Ri values within the PMSE. Not sure what is low and high, but it is hard to see if there really is more high or low values in the two colour panel. Is there a way to justify that statement? Reply: The summer mesopause region is characterized by a strong wind reversal from westward winds below to eastward winds above. This strong vertical shear leads to 'lower Ri' compared to the winter season. In winter there is almost no vertical shear in the mean winds (Wilhelm et al., 2017, Stober et al., 2017, Pokhotelov et al., 2018).

Comment: What are the GW-like periods you focus on in Figure 8? It seems like there is significantly less wave activity during the latter KHI event? Reply: The second KHI event did indeed occur under much less active GW conditions. The wavelet spectra in Figure 6 indicate a significant activity for short period waves for the occurrence time of the second GW. However, we cannot provide information about the source of the gravity waves.

Comment: Line 8 on page 5 says:"The zonal and meridional winds are dominated by the tides." A sentence saying how the data plot supports this idea would be helpful. Reply: The statement is based on the continuous meteor radar observations. We consider periods around 24 h, and 12 hours as tides, if they only show a weak day-to day variability. Gravity waves should not have every day similar periods and occurrence time, they are supposed to have a more intermittent behavior.

Comment: When Richardson number is being introduced in section 4.1, a brief reasoning for why that is a useful parameter in the KHI study would be good. Reply: Numerical simulations of shear flows show that KHI can evolve when the Ri<0.25. (see Fritts et al., ) and many other authors. This is also introduced in the manuscript.

Comment: Can you specify the meaning of "coherent wave-like structure" (page 7, line 23)? What is coherent enough? Reply: All pictures/images of the 2D scans were search by hand to identify what we consider a coherent structure (as shown in Figure

9). However, we did not introduce yet a coherency in a strict mathematical sense. Once we found, what we call coherent, we applied a non-linear fitting and only if we achieved a convergence of the fit we kept the results.

Comment: What is the significance or implication of the prevailing wind direction with respect to the ripple propagation direction? The observations and the plot are not really discussed. Reply: Linear theory predicts that GW can only propagate as long as their phase speed is different than the background wind. As most of the observed ripples travel against the mean flow they have rather low observed phase speeds, but due to the strong mean winds they have high intrinsic phase speeds. It appears that these waves are significantly Doppler shifted in their observed frequencies. However, when the ripples evolve into KHI they stop propagating and start to be advected with the wind velocity (private communication Dave Fritts). In addition our technique has an observational filter that emphasis such fast propagating waves moving against the mean flow.

Comment: When introducing earlier observations by Demissie et al. (page 9, line 12), you mention that those are from different years. Does that mean that you would expect annual differences? Reply: We mentioned explicitly the different years to avoid potential mistakes that the same waves could have been observed. So we wanted to clarify that both studies observed entirely different GW events and so only the statistical properties could be compared. However, we expect also some year to year differences depending on the tropospheric synoptic situation.

Minor comments: Comment: The introduction mentions "mesoscale" many times. It is a relative measure which depends on the observations, so it would be good to give a rough number or range for it. Reply: The term mesoscale is a defined meteorological horizontal scale covering structures larger than 5 km up to several thousand of kilometer (size of synoptic weather pattern).

Comment: The paragraph change on page 1 and line 23–24 is unnecessary when

the first sentence of the new paragraph refers to the last sentence of the previous paragraph. Reply: We removed the paragraph.

Comment: On page 2 and line 9, "and many other" is redundant since the reference list starts with "e.g.". Reply: Changed.

Comment: On page 3 and line 21, Figure 2 is hardly a contour plot. Reply: Figure 2 consists of several panels. The upper two are contour plots of the radial velocity and the statistical uncertainty. The lower two panels are a histogram of the statistical errors of the radar velocities and a scatter plot of SNR vs. radial velocity statistical uncertainty.

Comment: On page 4 and line 4, should the "zenith distance" be a zenith angle since the rest of the sentence talks about degrees? Reply: Changed.

Comment: Figured 3 has an "a)" as a panel marking but I do not see the panel labels b and c. Reply: The label a) is removed.

Comment: The description of Figure 5 in the text says that there is Ri calculated from MAARSY and MR data (the plural s in "lower panels" seems redundant). The figure caption says that the panels are Ri from wind shear and Brunt-Väisälä frequency. Could you add the data source information in the figure caption to make it more self-explanatory? Reply: We expanded the caption of the figure and provide the suggested information.

Comment: Figure 3 does not show any red boxes but based on the statement on the top of page 6, it might make a difference to generate the boxes. Reply: We removed the mentioning of red boxes from the text.

Comment: What is DNS simulation (page 6, line 9)? Reply: We added to the text (DNS-direct numerical simulation).

Comment: Does "rather common" (page 6, line 14) mean "not uncommon", or is there actually a description attached to it, which could be added to be a bit more precise? Reply: Yes we are using rather common in the sense of not uncommon. However, we

cannot provide a statistical analysis of how often such structures are seen in airglow images. This is beyond the scope of this paper, but maybe such events are more often published than other images.

Comment: On page 6 and line 24: "than" should probably be "then" Reply: Corrected.

Comment: "Train of ripples" referring to Figure 7a seems right, but in Figure 7b it looks more like one single wave-like feature. Reply: We clarified this in the text.

Comment: Should be "wave-like" instead of "wave.like" on page 8 and line 16. Reply: Corrected.

Comment: On page 9 and line 8 the text blames weather conditions for not having other observations. Does that relate to a lack of optical observations due to the daylight conditions, or does it really mean weather? Reply: Actually both effects are relevant for the location at Andenes. To emphasis both effects we added 'clouds' and 'daylight' at some passages in the text.

Comment: Seems like there is an extra "propagation direction" on page 10 and line 9. Reply: Corrected.
* * *